# Impact of Multiplex PCR in the Therapeutic Management of Severe Bacterial Pneumonia

**DOI:** 10.3390/antibiotics13010095

**Published:** 2024-01-18

**Authors:** Julien Dessajan, Jean-François Timsit

**Affiliations:** 1Assistance Publique Hôpitaux de Paris (AP-HP), Medical and Infectious Diseases Intensive Care Unit, Bichat Claude-Bernard Hospital, Paris Cité University, 46 Rue Henri Huchard, 75018 Paris, France; julien.dessajan@aphp.fr; 2Mixt Research Unit (UMR) 1137, Infection, Antimicrobials, Modelization, Epidemiology (IAME), Institut National de la Recherche Médicale (INSERM), Paris Cité University, 75018 Paris, France

**Keywords:** rapid tests, sepsis, nosocomial pneumonia, ICU, diagnosis, resistance

## Abstract

Pneumonia is a common and severe illness that requires prompt and effective management. Advanced, rapid, and accurate tools are needed to diagnose patients with severe bacterial pneumonia, and to rapidly select appropriate antimicrobial therapy, which must be initiated within the first few hours of care. Two multiplex molecular tests, Unyvero HPN and FilmArray Pneumonia+ Panel, have been developed using the multiplex polymerase chain reaction (mPCR) technique to rapidly identify pathogens and their main antibiotic resistance mechanisms from patient respiratory specimens. Performance evaluation of these tests showed strong correlations with reference techniques. However, good knowledge of their indications, targets, and limitations is essential. Collaboration with microbiologists is, therefore, crucial for their appropriate use. Under these conditions, and with standardized management, these rapid tests can improve the therapeutic management of severe pneumonia faster, more precisely, and with narrow-spectrum antibiotic therapy. Further randomized controlled trials are needed to address the many unanswered questions about multiplex rapid molecular testing during the diagnosis and the management of severe pneumonia. This narrative review will address the current knowledge, advantages, and disadvantages of these tests, and propose solutions for their routine use.

## 1. Introduction

Severe pneumonia remains a major cause of morbidity and mortality worldwide, and its therapeutic management is a public health issue. The severity of pneumonia is generally defined by clinical criteria leading to admission to an intensive care unit (ICU) [1]. One of them is the need for mechanical ventilation (invasive or non-invasive), or severe hypoxemia defined by a PaO2/FiO2 ratio of less than 300 mmHg, requiring oxygen administration through a high-flow nasal cannula or a non-rebreathing mask. Among hospital-acquired pneumonia (HAP), the most severe and frequent is the ventilator-associated pneumonia (VAP). It is defined as an infection of the lung parenchyma in patients receiving invasive mechanical ventilation for at least 48 h. Conversely, community-acquired pneumonia (CAP) refers to episodes in patients with no recent healthcare exposure. In Europe, the estimated incidence of VAP is 18.3 episodes per 1000 ventilator-days [2,3]. In-ICU mortality attributable to VAP is limited but significant (from 1 to 6% according to case mix and methods), suggesting that mortality in these patients is mainly driven by their underlying conditions, as well as the severity of the disease [3,4,5]. However, VAP has been frequently associated with a longer duration of mechanical ventilation, ICU stay, prolonged hospitalization, and increased healthcare cost [4].

### 1.1. Basic Rules for Treatment Choice

Adequate initial antibiotic therapy has proven to be significantly associated with improved survival in severe CAP and VAP [6,7]. Management of these patients relies on the early introduction of empirical antibiotic therapy after respiratory sampling. The selection of antibiotics for empirical therapy depends above all on the context of the infection.

Knowledge of CAP, HAP, and VAP epidemiology, as well as adherence to guidelines, are the basic rules for treatment selection. For hospital-acquired cases, it is necessary to consider the local epidemiology, previous colonization, and any ongoing outbreaks. Physicians also need to consider traditional risk factors for multi-drug resistant (MDR), extensive drug resistant (XDR), and Difficult-to-Treat resistant (DTR) bacteria, such as length of stay, acute respiratory distress syndrome (ARDS), or shock. The initial antimicrobial choice also depends on previous antimicrobial therapy. Despite knowledge of the risk factors for bacterial resistance, empirical therapy was often inadequate, particularly for infections caused by DTR bacteria [8]. In CAP, unnecessarily broad-spectrum antimicrobial therapy has also been shown to be associated with increased mortality, longer length of stay, and higher costs when compared to narrow-spectrum antimicrobial therapy [9].

Our aim is, therefore, to have an antibiotic therapy that is immediately appropriate. Conversely, antibiotics should not be administered when not needed in order to minimize adverse events and the development of resistance.

### 1.2. Current Specific Needs According to Guidelines

A microbiological diagnosis of pneumonia is notably difficult to obtain. Only an estimated 38% of hospitalized CAP cases are microbiologically documented [10]. The main reasons for these results are difficulties in obtaining deep sputum samples, antibiotic therapy often started before sampling, low sensitivity of the techniques used, and non-detection of certain pathogens.

To improve the microbiological diagnosis of pneumonia, new molecular rapid tests have been developed using the mPCR technique. These tests allow rapid microbiological diagnosis, and guide the initiation of an appropriate antibiotic therapy, its duration, escalation and/or de-escalation, and even its discontinuation if not necessary. From a more collective perspective, these tests can reduce the use of broad-spectrum antibiotic therapy, and contribute to reduce bacterial resistance. Finally, for certain pathogens, early documentation may allow patients to be isolated to prevent the spread of infection within the healthcare facility.

In the recent ERS/ESICM/ESCMID/ALAT guidelines for the management of severe CAP, experts suggest sending a lower respiratory tract sample for mPCR testing whenever non-standard severe CAP antibiotics are prescribed, or considered, with a very low quality of evidence [1]. For these patients, early and appropriate therapy is required, depending on the causative pathogen suspected. For CAP caused by intracellular organisms, clinicians should optimize treatment by rapidly discontinuing inapropriate antibacterial agents such as beta-lactams. Early identification of *Staphylococcus aureus* and its resistance profile will help optimize treatment with an antitoxin or an agent active against methicillin-resistant *S. aureus* (MRSA). Pathogens present in specific populations, such as *Pseudomonas aeruginosa* or Enterobacterales, also require early, optimized targeted therapy. On the other hand, it is important to discontinue antibacterial agents early when they are not needed, such as viral CAP.

In HAP and VAP, the choice of early and appropriate therapy is crucial. Rapid diagnostic tools should therefore reduce the uncertainty of empirical treatment and enable prompt, appropriate treatment. In particular, new techniques based on rapid multiplex molecular diagnostic tests have recently been commercialized to improve the microbiological diagnosis of severe pneumonia. These tests are able to detect different viral and bacterial pathogens, allow semi-quantification of copies, and detect the micro-organisms present in the clinical sample and the targeted resistance genes within hours after sampling.

We will review the available data on their accuracy in severe pneumonia, as well as their impact on routine antimicrobial use, costs, and outcomes.

## 2. Multiplex PCR: Available Tests

Two US Food and Drugs Administration-approved CE-marked tests are currently available for pneumonia diagnosis. The Unyvero HPN (Curetis, Unyvero TM) detects 21 bacteria and one parasite, semi quantitatively (from + to +++), and identifies 15 resistance genes in approximately 5 h. The FilmArray Pneumonia+ Panel (BioFire, bioMérieux) detects 18 bacteria, among which are three atypical ones, quantitatively (from 10^4^ to ≥10^7^ genomic copies/mL), and identifies seven resistance genes, and eight viruses in approximately 90 min. The main targets of both tests are shown in the Table 1.

## 3. Biological Accuracy

### 3.1. General Comments

Prior to any description of the available data on the diagnostic performance of these new tests, it should be highlighted that some pathogens potentially involved in HAP or VAP are not represented in the panels. In particular, their contribution to the diagnosis of HAP or VAP is hampered by their lack of detection of *Hafnia alvei* and *Citrobacter koseri* for both panels, and of *Citrobacter freundii*, *Klebsiella variicola*, *Morganella morganii* and *Stenotrophomonas maltophilia* for the FilmArray Pneumonia+ panel. Importantly, if the micro-organism is not detected or under the quantification threshold, the result of bacterial resistance genes is then unavailable. Finally, the theoretical turnaround time is well known, but it needs to be integrated into laboratory activities and the routine results process to get the local turnaround time.

### 3.2. Qualitative Comparison of mPCR versus Standard-of-Care

#### 3.2.1. Viral and Bacterial Detection

Several studies have evaluated mPCR in pneumonia, revealing good diagnostic performances and high concordance with bacterial culture results. After reviewing bronchoalveolar lavage (BAL) specimens from 259 patients with suspected pneumonia, FilmArray Pneumonia+ Panel demonstrated a positive percentage agreement (PPA) of 96.2% and a negative percentage agreement (NPA) of 98.1% for the qualitative identification of 15 bacterial targets compared to routine bacterial culture. For viruses, the PPA has also done well (96.7%) as compared to monoplex PCR [11]. A recent meta-analysis including 30 observational studies and 8969 samples also revealed high diagnostic performance for the FilmArray Pneumonia+ Panel as compared to a standard culture [12]. The authors pointed out the lower diagnostic performance of mPCR for sputum, due to frequent contamination by oropharyngeal flora, leading to numerous false positive results.

Both available tests offer comparable diagnostic performances (Table 2). On 846 prospective BAL specimens, the sensitivity of the FilmArray Pneumonia+ panel remains above 90% for 15 commonly identified bacteria [13]. However, for the 1408 specimens evaluated by Unyvero HPN, PPA was 77.8% for *Enterobacter cloacae* complex and 89.1% for *Klebsiella pneumoniae* compared with the standard of care [14].

#### 3.2.2. Operative Values

We performed a literature review of studies comparing the mPCR film array with standard techniques. We used PubMed to select studies that compared the mPCR FilmArray Pneumonia+ Panel with the standard technique in cases of suspected bacterial pneumonia in adult patients. We used the following search terms: ‘respiratory infections’, ‘respiratory infections’, ‘respiratory infections’, ‘respiratory infections’, ‘respiratory infections’, ‘pneumonia, bacterial’, ‘bacterial pneumonias’ and ‘multiplex polymerase chain reaction’, ‘PCR, multiplex’, ‘multiplex PCR’. Of the 30 studies identified between 2019 and 2022, we selected 15 studies that included only patients with suspected bacterial pneumonia. Eight of the studies included patients admitted to ICU [15,16,17,18,19,20,21,22], and seven included patients from the wards, with even a few outpatients [23,24,25,26,27,28,29]. A total of 4596 samples were collected from 4204 patients, of which 43% were BAL (Table 3). When data were available, in 7 out of 15 studies, the authors reported that 46% of patients were treated with antibiotics prior to specimen collection.

It must be highlighted that, for these 15 pooled studies, 9% of the pathogens involved were absent from the FilmArray Pneumonia+ Panel. For bacterial identification by mPCR compared with standard culture, the sensitivity of these studies was 92% and specificity 97% (Table 4).

The turnaround time of mPCR depends on the laboratory’s organization and the availability of appropriate equipment. In a recent single-center prospective randomized study from Norway, the median observed turnaround time-to-results of the FilmArray Pneumonia+ Panel in suspected CAP [30] was shorter compared with sputum cultures (2.6 h vs. 57.5 h, *p* < 0.001).

The results of resistance detection based on the cumulative accounts of 15 studies deserve special attention. When the risk of resistance is low, the probability of missing out is also low. In this way, the PPA for resistance detection is around 95%, with a probability of missing it below 5% (Table 5). But, for these resistances which are rarely present, the risk of overdiagnosis is quite high (from 15 to 49%). Thus, the probability of true resistance with a positive mPCR is only 51% to 85%. These data highlight the risk of inappropriate therapeutic escalation in these patients.

Consequently, adequately interpreting the results is complex. The use of these results requires a specific training process, as well as close collaboration with microbiologists and infectious disease experts. The clinical impact on patient prognosis and antibiotic use is, therefore, still uncertain. It is of crucial importance to look to the results in view of the pretest probability of having a result. Even with a biologically accurate test, with sensitivity and specificity above 90%, the positive predictive value (i.e., the probability of having a resistance gene if the mPCR is positive for that resistance gene) is not maximized. As an example, if the probability of having a microorganism with blaOXA-48 is 0.2%, the mPCR will overdiagnose OXA-48 positivity in 37.5% of the cases (Table 5).

## 4. Clinical Impact: Published Works

Recent works have tried to determine the optimal place of mPCR in the diagnosis process. A first United Kingdom (UK) study evaluated mPCR from 323 adults with radiologically-confirmed CAP explored mainly by sputum specimens (96%) [31]. Molecular testing achieved pathogen detection in 87% of CAP patients compared with 39% with culture-based methods. Among patients who had received antimicrobials within the 72 h prior to admission (85%), the pathogen detection rate was also clearly higher with mPCR (78%) compared to standard culture (32%; *p* < 0.001). The authors concluded that molecular testing could have an impact on antibiotic prescribing, with de-escalation in 77% of patients; however, the study was neither controlled nor randomized.

A retrospective multicenter study was conducted in four French university hospitals on 159 pneumonia episodes (HAP, CAP or VAP), with 81% of patients hospitalized in ICUs [32]. On the basis of the mPCR results, the multidisciplinary committee comprising an intensivist, an infectiologist, and a clinical microbiologist proposed a change in empirical treatment in 77% of cases. In microbiologically documented episodes, mPCR increased the appropriateness of empirical treatment to 87%, compared with 77% in routine care.

Our French team simulated the impact of mPCR on 95 clinical samples from 85 ICU patients with VAP (75%) and ventilated HAP (25%) [33]. Simulated antimicrobial strategy was compared in two different groups: one group with medical history and Gram staining results, and the other group with the same data and mPCR results processed by a panel of experts. In this prospective study, mPCR could led to antibiotic changes in 63/95 (66%) episodes of pneumonia with early initiation of effective antibiotic in 20/95 (21%) patients and early de-escalation in 37/95 patients (39%). However, mPCR could also have led to one (1%) inadequate antimicrobial therapy. Among 17 empiric antibiotic treatments with carbapenems, 10 could have been de-escalated in the following hours according to the mPCR results. Thus, the simulated impact observed in this prospective study appears promising, but needs to be confirmed by randomized controlled trials.

A single-center randomized controlled trial was conducted in UK on 200 critically ill adults with pneumonia (CAP 42%, HAP 35% and VAP 23%) [34]. Patients were allocated (1:1) to mPCR combined with an antibiotic stewardship strategy, or to routine clinical care. Eighty (80%) of patients in the interventional group received results-directed therapy, which was the primary outcome, compared with 29 (29%) of 99 in the control group (difference of 51%, 95% CI 39–63; *p* < 0.0001). In the mPCR group, 42% of patients had antibiotics de-escalated compared with 8% in the control group. Despite these major differences in therapeutic strategy, there were no major differences in clinical results or safety between the two groups. The authors therefore concluded that mPCR was associated with improvements in antimicrobial use and appeared to be safe.

The Flagship II was a multicenter, randomized controlled trial conducted in Switzerland. Patients with high risk of Gram-negative bacteria CAP and HAP (n = 208) proven by BAL performed by bronchoscopy were randomly assigned (1:1) to standard care or mPCR followed by antibiotic stewardship recommendation within five hours of invasive sampling [35]. A 45% reduction in the duration of inappropriate therapy was observed. Inappropriate therapy was defined as too broad, too long, or inadequate. However, there was no effect on clinical stability, antibiotic-related adverse events (5%), length of stay, or mortality (8%). The external applicability of this study may be compromised by its design. Routine bronchoscopy for all patients does not correspond to current worldwide practice. Furthermore, the study design does not allow to discriminate the effect of the mPCR itself from the antibiotic stewardship.

In the particular context of the recent COVID-19 pandemic, another study attempted to evaluate an antibiotic de-escalation algorithm based on the combination of mPCR and procalcitonin (PCT) results in 194 patients who were critically ill with SARS-CoV-2 pneumonia [36]. As expected, respiratory bacterial co-infection rate was higher in mPCR group (45/93, 48.4%) than in standard-of-care group (21/98, 21.4%). The authors were unable to demonstrate a reduction in overall antibiotic exposure or a benefit in terms of clinical outcomes at day 28. These disappointing results from the intention-to-treat analysis could be partly explained by significant protocol deviations during this study, which was conducted in an exceptional context. Indeed, in the per-protocol analysis, the number of antibiotic-free days after randomization was two days higher at day seven in the intervention group than in the control group (4 days vs. 2 days; RR 1.38 (1.01 to 1.88)). Unfortunately, the antibiotics were reintroduced, and the results for antibiotic-free days on day 28 were no longer significant (14 days vs. 15 days; RR 0.98 (0.66 to 1.46)).

The INHALE trial, a multicenter, randomized controlled trial conducted in 13 UK ICUs, was presented at IDWeek in November 2022 [37]. The aim was to evaluate antimicrobial stewardship improvement though implementing mPCR in 556 patients admitted for HAP or VAP. The authors compared standard-of-care with a strategy involving mPCR combined with a specific antibiotic algorithm. The antimicrobial appropriateness at 24 h was significantly better in the intervention group compared to the control group (76.5% vs. 55.9%, *p* < 0.001), and this benefit remained significant at 72 h (73.4% vs. 58.8%, *p* < 0.001). mPCR groups failed to demonstrate non-inferiority in term of clinical cure of pneumonia at 14 days. It was 56.7% in the intervention group and 64.7% in the control group (95% CI −0.15 to 0.02), with exploratory analyses suggesting site heterogeneity. The cause of this failure is difficult to determine, and may be due to the lack of accuracy of the mPCR, the de-escalation algorithm, or the investigator’s failure to follow the algorithm.

A randomized controlled trial was recently conducted in three Danish emergency departments in patients with CAP [38]. The FilmArray Pneumonia+ Panel was compared with standard care on non-invasive specimens, including tracheal secretion (78.4%) or sputum (21.6%), from 294 patients. No difference was found between the two groups in the primary outcome of prescriptions of no or narrow-spectrum antibiotics at 4 h after admission (62.8% in mPCR group and 59.6% in standard of care group, *p* = 0.134). Based on patients with positive culture results (n = 55), secondary outcomes showed that prescriptions in the mPCR group were more targeted at 4-h (OR 5.68, 95% CI [2.49, 12.94]) and 48-h (OR 4.20, 95% CI [1.87, 9.40]) and more appropriate at 48-h (OR 2.11, 95% CI [1.23, 3.61]). However, the authors did not find difference in terms of 30-day mortality (OR 0.90; (95% CI [0.43, 1.86] *p* = 0.787) or transfer to ICU (OR 0.54, 95% CI [0.10, 2.91] *p* = 0.475), although the number of events was very low.

## 5. Perspectives

Several studies are underway to better assess the impact of these molecular tests on patient outcomes. MULTI-CAP, a randomized controlled trial focusing on severe CAP, aims to evaluate the effectiveness of a management strategy combining mPCR and a de-escalation and early antibiotic discontinuation algorithm based on both mPCR and PCT results [39]. A second trial, SHARP (NCT04153682), will evaluate the impact on antibiotic strategy of mPCR in addition to standard of care in patients with HAP or ventilated HAP. Both studies are completed, and results will be soon available. RESPIRE (NCT05405491) will test the added value of mPCR to optimize antibiotics in immunocompromised patients with HAP requiring mechanical ventilation.

Many questions remain to be answered concerning the use of mPCR. In fact, in routine practice, the clinician’s decision should be based not only on the mPCR result, but also on many other parameters. First, the context: as mentioned above, the rationale for CAP privileges the appropriate choice of antibiotic, and avoidance of antibacterial therapy when the etiology is viral. In contrast, in the context of HAP/VAP, the clinician will decide on empiric therapy according to the intensity of the clinical deterioration. If empirical antibiotic therapy is adopted, the choice of molecules will depend on many factors such as the previous ecology of the unit, previous bacterial colonization of the patient, previous use of antimicrobial therapy, and Gram stain examination. The mPCR result must be interpreted in the light of all these factors which determine the pre-test probabilities of pathogens and antimicrobial resistance. Importantly, interpretation of mPCR results requires knowledge of the resistance mechanisms for each microorganism in your center. Key elements to know are that mPCR is not able to diagnose ampC production, impermeability or efflux pump mechanisms of resistance. These undetected mechanisms are common for example in *Pseudomonas aeruginosa* and group 3 Enterobacterales [40,41,42]. A routine algorithm representing our decision tree for HAP and VAP at Bichat Hospital (Paris, France) is depicted in Figure 1.

First, the significance of a positive mPCR in the presence of a negative culture needs to be clarified. As previously described, the use of mPCR can lead to the unnecessary introduction or escalation of antibiotic treatment. If appropriate antibiotic therapy is started promptly prior to respiratory sampling, some bacterial cultures may rapidly turn out to be negative [43]. The impact of pre-sampling antimicrobial therapy on mPCR results remains unknown. Only one single-center study suggests that discordance between mPCR and culture results are more important in the presence of previous antimicrobial therapy [44]. The use of mPCR as a rescue tool for the microbiological diagnosis of pneumonia whose sample has been negative due to early treatment remains to be explored.

The second major area to explore is the role of DNA copy number quantification. Only a poor correlation, ranging from 40% to 56%, was found between the number of DNA copies obtained by mPCR and the colony-forming units (CFU) of bacterial cultures [45]. Consequently, there is no consensus on their use in clinical practice. The use of a universal threshold, for example to distinguish colonization from infection, is therefore not yet a practical option. Furthermore, the evolution of mPCR results during appropriate antibiotic treatment is not known. Its use as a marker of microbiological cure has therefore never been evaluated.

Some preliminary study suggests, for both CAP and HAP, that viral-bacterial coinfection may have prognostic impact not only in influenza or SARS-CoV2 infections but also for other respiratory viruses [46,47,48]. Finally, the consequences of systematic screening of a viral co-infection in mPCR has not been evaluated either.

## 6. Conclusions

Multiplex PCR appears as a valuable tool in the management of severe bacterial pneumonia, as it is easy to perform, rapid and sensitive. However, there are some limitations for its use. First, it should not be used alone, and a prior knowledge of the product is required. Appropriate training, compliance with international recommendations and expert help are essential for the proper use of these tests. As always, it is essential to consider the pre-test probability of a particular etiology and resistance pattern in order to make an appropriate decision. Before performing these tests, the hypothesis and the question asked must be clearly stated in order to optimize the interpretation of the results.

In addition, as described above, the ongoing trials are still struggling to show a strong and clinically significant effect. At least partly because it’s a new tool, the technical aspects are not described exhaustively. The interpretation and evolution of copy number per milliliter under treatment, or the identification of a clinically relevant threshold, are areas that still need to be explored. In addition, it may be appropriate to limit the number of targets to a specific panel for each clinical situation, for example the CAP versus HAP panel or the immunocompetent versus immunocompromised patient panel.

Finally, clinical trial results are difficult to transpose to real-life situations, as they are produced by trained expert teams. Education and training of clinicians in microbiology are therefore fundamental for the adoption of these new techniques and their integration into routine practice.

## Figures and Tables

**Figure 1 antibiotics-13-00095-f001:**
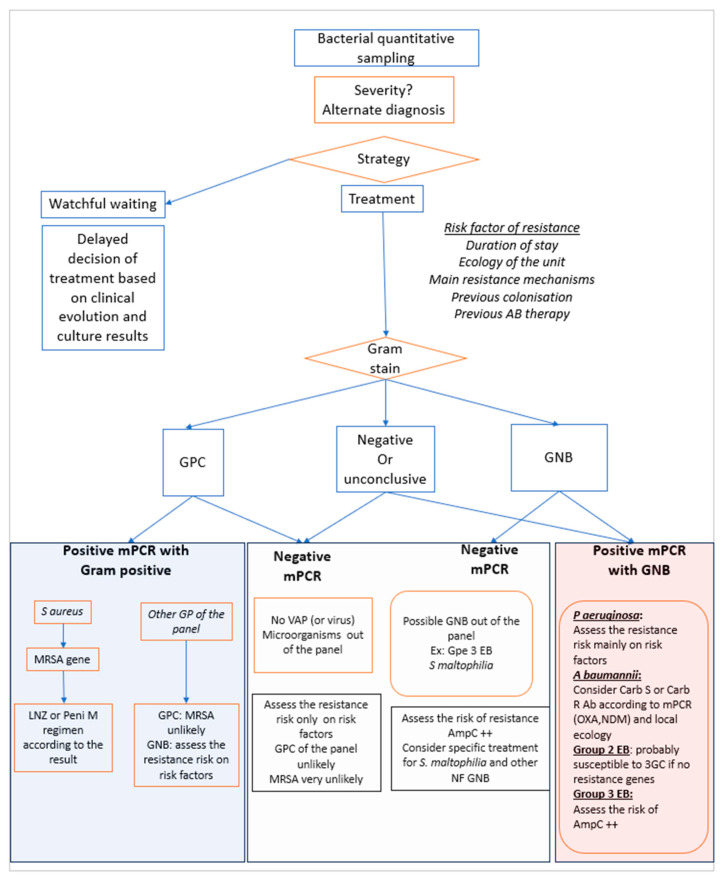
Proposed decision algorithm that integrates mPCR results in the choice of empirical therapy in HAP and VAP. AB: antibiotic, GPC: Gram positive cocci, GNB: Gram negative bacilli, mPCR: multiplex polymerase chain reaction, LNZ: linezolid, Peni: Penicillin, MRSA: methicillin-resistant *Staphylococcus aureus*, VAP: ventilatory-associated pneumonia, NF GNB: non fermentative Gram-negative bacilli, Carb: carbapenem, OXA and NDM: carbapenemases gene. 3GC: third generation cephalosporins.

**Table 1 antibiotics-13-00095-t001:** Unyvero HPN and FilmArray Pneumonia+ Panel main targets.

	Unyvero HPN	FilmArray Pneumonia+ Panel
Number of targets	36	33
Turnaround time	5 h	90 min
Type of detection	Semiquantitative (+ to +++)	Quantitative (10^4^ to ≥10^7^)
Included pathogens		
**Bacteria**		
**Gram-positive cocci**		
*Staphylococcus aureus*	x	x
*Streptococcus agalactiae*		x
*Streptococcus pneumoniae*	x	x
*Streptococcus pyogenes*		x
**Gram-negative cocci**		
*Moraxella catarrhalis*	x	x
**Gram-negative bacilli**		
*Haemophilus influenzae*	x	x
Group 1 Enterobacterales		
*Escherichia coli*	x	x
*Proteus* spp.	x	x
Group 2 Enterobacterales		
*Klebsiella oxytoca*	x	x
*Klebsiella pneumoniae*	x	x
*Klebsiella variicola*	x	
Group 3 Enterobacterales		
*Enterobacter cloacae complex*	x	x
*Citrobacter freundii*	x	
*Enterobacter cloacae complex*	x	x
*Klebsiella aerogenes (Enterobacter aerogenes)*	x	x
*Morganella morganii*	x	
*Serratia marcescens*	x	x
Non-fermenting bacteria		
*Acinetobacter baumannii complex* *	x	x
*Pseudomonas aeruginosa*	x	x
*Stenotrophomonas maltophilia*	x	
**Atypical bacteria**		
*Chlamydia pneumoniae*	x	x
*Legionella pneumophila*	x	x
*Mycoplasma pneumoniae*	x	x
**Others**		
*Pneumocystis jirovecii*	x	
**Resistance genes**	x	x
**Virus**	x	x

* *Acinetobacter calcoaceticus-baumannii* complex for FilmArray Pneumonia+ Panel.

**Table 2 antibiotics-13-00095-t002:** Unyvero HPN and FilmArray Pneumonia+ Panel performance compared to SoC, by Klein, M. et al. and Murphy, C.N. et al. [13,14].

Species	No. Positive by Unyvero and SoC/No. Positive by SoC	Unyvero PPA (%)	No. Positive by FilmArray and SoC/No. Positive by SoC	FilmArray PPA (%)
*Acinetobacter baumannii complex* *	28/29	97	10/11	91
*Citrobacter freundii*	6/6	100	ND	
*Enterobacter cloacae complex*	28/36	78	22/24	92
*Escherichia coli*	63/67	94	35/38	92
*Haemophilus influenzae*	58/59	98	26/28	93
*Klebsiella oxytoca*	22/24	92	11/11	100
*Klebsiella pneumoniae*	49/55	89	36/38	95
*Moraxella catarrhalis*	23/23	100	5/5	100
*Proteus* spp.	19/19	100	20/20	100
*Pseudomonas aeruginosa*	128/128	100	139/142	98
*Staphylococcus aureus*	119/129	92	157/159	99
*Serratia marcescens*	35/37	95	32/33	97
*Stenotrophomonas maltophilia*	56/61	92	ND	
*Streptococcus pneumoniae*	37/38	97	21/21	100

SoC: Standard of Care i.e., quantitative reference culture, ND: not detected, PPA: positive percentage agreement, * *Acinetobacter calcoaceticus-baumannii* complex for FilmArray Pneumonia+ Panel.

**Table 3 antibiotics-13-00095-t003:** Operative value of 15 studies comparing mPCR (FilmArray Pneumonia+ Panel) performance to SoC in patients with suspected bacterial pneumonia, general data [15,16,17,18,19,20,21,22,23,24,25,26,27,28,29].

Author (Year)	No. Patients	Center	No. Samples	BAL (%)	Antibiotic before Sampling (%)	Out of the Panel (%)
Edin, A. et al. (2020) [23]	84	ward	84	19	NA	14
Crémet, L. et al. (2020) [15]	100	ICU	237	32	25	8
Kolenda, C. et al. (2020) [16]	99	ICU	99	49	72	NA
Lee, S.H. et al. (2019) [17]	51	ICU	59	32	NA	31
Caméléna, G. et al. (2021) [18]	43	ICU	96	100	67	6
Mitton, B. et al. (2021) [24]	59	ward	59	2	NA	7
Gastli, N. et al. (2021) [25]	515	ward	515	47	NA	9
Foschi, C. et al. (2021) [19]	178	ICU	230	23	NA	6
Maataoui, N. et al. (2021) [22]	67	ICU	112	94	79	10
Kyriazopoulou, E. et al. (2021) [26]	90	other	90	0	0	0
Ginocchio, C.C. et al. (2021) [27]	2463	NA	2463	50	NA	13
Posteraro, B. et al. (2021) [20]	150	ICU	212	39	28	1
Kayser, M.Z. et al. (2022) [28]	60	other	60	100	50	7
Fontana, C. et al. (2021) [29]	152	ward	152	43	NA	4
Cohen, R. et al. (2021) [21]	93	ICU	128	19	NA	11
TOTAL	4204		4596	43	46	9

Results related to viral pathogens are not reported. NA: not applicable, SoC: Standard of Care i.e., quantitative reference culture.

**Table 4 antibiotics-13-00095-t004:** Operative value of 15 studies comparing mPCR (FilmArray Pneumonia+ Panel) to SoC in patients with suspected bacterial pneumonia, test performance [15,16,17,18,19,20,21,22,23,24,25,26,27,28,29].

Author (Year)	TP	FN	FP	TN	Sensitivity (%)	Specificity (%)
Edin, A. et al. (2020) [23]	39	2	26	1118	95	98
Crémet, L. et al. (2020) [15]	194	2	200	3159	99	94
Kolenda, C. et al. (2020) [16]	16	0	26	1443	100	98
Lee, S.H. et al. (2019) [17]	27	3	17	838	90	98
Caméléna, G. et al. (2021) [18]	36	2	8	1394	95	99
Mitton, B. et al. (2021) [24]	46	4	52	783	92	94
Gastli, N. et al. (2021) [25]	374	22	294	7035	94	96
Foschi, C. et al. (2021) [19]	86	10	57	3297	90	98
Maataoui, N. et al. (2021) [22]	43	9	5	1623	83	100
Kyriazopoulou, E. et al. (2021) [26]	7	1	100	1242	88	93
Ginocchio, C.C. et al. (2021) [27]	1661	127	1371	33786	93	96
Posteraro, B. et al. (2021) [20]	180	0	22	2978	100	99
Kayser, M.Z. et al. (2022) [28]	19	5	0	876	79	100
Fontana, C. et al. (2021) [29]	45	5	98	2132	90	96
Cohen, R. et al. (2021) [21]	83	7	61	1769	92	97
TOTAL	2856	199	2337	63473	92	97

Results on viral pathogens are not reported. SoC: Standard of Care i.e., quantitative reference culture, TP: true positive, FP: false positive, FN: false negative, TN: true negative.

**Table 5 antibiotics-13-00095-t005:** Detection of resistance based on cumulative counts from 15 studies comparing mPCR (FilmArray Pneumonia+ Panel) to SoC in patients with suspected bacterial pneumonia [15,16,17,18,19,20,21,22,23,24,25,26,27,28,29].

Antimicrobial Resistance Gene	Rate (%)	TP	FN	FP	TN	No. Positive mPCR	Resistance	
							Missed (%)	Overdiagnosed (%)
mecA/mecC and MREJ	3%	54	3	52	1830	106	5.2	49
CTX-M	4%	76	5	29	1829	105	6.1	27.6
NDM	0.5%	11	0	2	1926	13	0	15
OXA48-like	0.2%	5	0	3	1931	8	0	37.5

SoC: Standard of Care i.e., quantitative reference culture, TP: true positive, FP: false positive, FN: false negative, TN: true negative, mecA/mecC and MREJ: methicillin resistance (*S. aureus*), CTX-M: extended spectrum beta-lactamase, NDM and OXA48-like: carbapenemases.

## Data Availability

Not applicable.

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
