# Peer review of "Impact of Multiplex PCR in the Therapeutic Management of Severe Bacterial Pneumonia"

_antibiotics, 2024, doi:10.3390/antibiotics13010095_

Round 1
Reviewer 1 Report
Comments and Suggestions for Authors
Nice work, just few suggestions for you to consider.

Comments on the Quality of English LanguageNone
Author Response
Manuscript ID: Antibiotics-2818766
Title: Impact of multiplex PCR in the therapeutic management of severe pneumonia.
General Comments – The manuscript is well written, nicely organized, and easy to follow. I just have few minor comments for the authors to consider. Not sure if it is important to specify which mPCR was used in each study (i.e., Unyvero HPN or FilmArray Pmeumonia +Panel) and if one is better than the other.
Done: We only mentioned it for data selection accuracy. Also for interventional study it might be of interest since the turn-around time are different between techniques.
Abstract:
Line #14 – Spell out PCR here “polymerase chain reaction”. I know it is a well-known abbreviation, but I am sure all abbreviations should be spelled out/defined first time then the abbreviated version used. Done
Line #15 – Should it be from “patient’s” respiratory specimens. Done
Line #17 – Since we are discussion two tests, should it be “knowledge of their indications”? Done
Line #18 – May consider changing the second “essential” to “crucial” or just “necessary” to avoid the duplication. Done
Introduction:
Line #44 – Consider changing “been shown” to “has proven”. Done
Line #57 – Delete the (.) just after bacteria. Done
Line #71 – A short sentence. Can easily delete. Done
Line #72 – Start the sentence with: These tests will provide rapid microbiological diagnosis, initiation of appropriate antibiotic therapy, duration, escalation and/or de-escalation, and even discontinuation if not necessary. Done
Biological Accuracy
Line #123 – Consider changing “virus” to “viral”. Done
Line #133 – Consider changing “found” to “identified” bacteria. Done
Line #145 – Consider changing to 15 studies “were” identified between 2019 and 2022. Done
Line #145 – Line #147 – Consider the following minor revision for better flow: Eight of the studies included patients admitted to ICU, seven included patients from the wards, with even a few outpatients. Done
Line #169 – I think “appropriate equipment availably” may sound better. Done with “availability of appropriate equipment”.
Line #174 – Line #177 – Please review these few lines, I think something is missing; () did not close. Done
Clinical Impact: published works
Line #195 – Consider inserting (UK) after United Kingdome since you have used this abbreviation later. Done
Line #199 – Delete “difference”, just say the pathogen detection rate was or (rates were) also clearly higher with mPCR…. Done
Line #200 – I think it should be the authors “concluded”. Done
Monard et al. Study (30)
Line #204 – You may abbreviate “ICU” here since it has already been done. Should you mention 159 pneumonia episodes (HAP, CAP, VAP) with 81% of patients hospitalized in ICUs. Done
Poole et al. Study (32)
Line #220 – Line # 222 - Not sure why you have “first” here again. Is it because this study was randomized, controlled, etc. You may just say: a single-center randomized controlled trial was conducted in UK on 200 critically ill adults with pneumonia. Patients were assigned …. Done
Line #228 – The authors “concluded” as well as “appeared” Done
Darie et al. Study (33)
In the study starting Line #230 – I think it will be nice to say the Flagship II was a multicenter, randomized controlled trial in Switzerland. High risk CAP and HAP patients (n=208) proven by a BAL performed by bronchoscopy were randomly assigned (1:1) to mPCR or conventional … Done
Line #233 – What/who was the phone call about? Done replaced by « antibiotic stewardship recommendation”
Fartoukh M. et al. Study (34)
This is an interesting study and I think you should also include number of patients and that daily procalcitonin (PC) results since there has been controversy on its use, but it is still included in lab results in many centers. The study also mentioned the antibiotic discontinuation based on the PC results. Done
Line #243 – Change ”are” to “were”. Done
Line #250 – Correct 2-days. Done
High et al. Study (35)
Line #254 – Consider saying finally the INHALE trial, a multicenter, randomized controlled trial in 13 UK ICUs (okay to use the abbreviations since it has already been done so) which was presented at IDWeek in November 2022. The rest of the paragraph is good. Done
Perspectives
Line #267 – May consider saying: currently several trials are in progress to better assess ….One randomized controlled trial is… The rest of the paragraph sounds good. Done
Line #287 – “in your center” or “in your practice”. Done “in your center”
Line #307 – Change “negatived” to “negative”. Done
Conclusions
Line #322 – Should “conditions” be ‘limitations”, i.e., there are a few limitations for its use. Done
Line #329 – Consider changing “latest” to “the ongoing trials”. Done
Line #334 – I think “may also be appropriate” is extra. Consider deleting. Agreed. Done
Line #339 – Consider changing “routine care” to “routine practice”. Done
Reviewer 2 Report
Comments and Suggestions for Authors
It is a very well done article with lots of indications of the rapid tests that can improve the therapeutic management of severe pneumonia with faster, more precise spectrum antibiotic therapy, and can clarify many problems in diagnosis of severe pneumonia.
Author Response
thank you
the articles was reviewed for typo and grammatical errors.
Reviewer 3 Report
Comments and Suggestions for Authors
Dear Editor
The manuscript "Impact of multiplex PCR in the therapeutic management of severe pneumonia" is well written regarding the global health concern associated with different kind of infectious pneumonia. However, there are some of the minor suggestions to improve the manuscript.
It is suggested to describe the inclusion/ selection criteria of "studies" in abstract or in the first paragraph of the introduction.
The first 3 lines of the abstract needs to be attractive for the reader, Authors are suggested to improve the text of "Line 10-13".
Line 18: "microbiology team or experts" needs to be rephrased as "Collaboration with the microbiologist is therefore essential for developing an appropriate therapeutic strategy".
Line 21: the sentence could be started as "In addition to the available studies, further----------".
Line 40-41: it is suggested to also add the definition of CAP (6-7) in this paragraph.
Line 71: "The objectives of this test are multiple". This line seems to be incomplete. The authors are suggested to include 2-3 objectives here. or delete the statement.
Line 86: Staphylococcus can be written as S. aureus as it is already described in the text.
Line 105: from 104 to ≥ 107 needs correction and units as CFU/mL.
Line 210-219, Line 221-229, Line 232-240, Line 243-250: it is suggested to add the references here.
At the end the authors are requested to critically read the manuscript for corrections of minor English mistakes. Thanks and Regards
Thanks and Regards
Comments on the Quality of English LanguageThe quality of English language needs to be improved according to US/ UK standards. However, the European style is prominent in this manuscript.
Author Response
The manuscript "Impact of multiplex PCR in the therapeutic management of severe pneumonia" is well written regarding the global health concern associated with different kind of infectious pneumonia. However, there are some of the minor suggestions to improve the manuscript.
Done
The first 3 lines of the abstract needs to be attractive for the reader, Authors are suggested to improve the text of "Line 10-13". Agree. Done
Line 18: "microbiology team or experts" needs to be rephrased as "Collaboration with the microbiologist is therefore essential for developing an appropriate therapeutic strategy". Done with crucial instead of essential to avoid the duplication
Line 21: the sentence could be started as "In addition to the available studies, further----------". Done
Line 40-41: it is suggested to also add the definition of CAP (6-7) in this paragraph. Done
Line 71: "The objectives of this test are multiple". This line seems to be incomplete. The authors are suggested to include 2-3 objectives here. or delete the statement. Agreed we deleted it.
Line 86: Staphylococcus can be written as S. aureus as it is already described in the text. Done
Line 105: from 104 to ≥ 107 needs correction and units as CFU/mL. Done
Line 210-219, Line 221-229, Line 232-240, Line 243-250: it is suggested to add the references here.
Done
At the end the authors are requested to critically read the manuscript for corrections of minor English mistakes. Thanks and Regards
Thanks and Regards
Comments on the Quality of English Language
The quality of English language needs to be improved according to US/ UK standards. However, the European style is prominent in this manuscript.